# Bullying Perpetration and Homophobic Teasing: Mediation through Gender Role Attitudes

**DOI:** 10.3390/children9081127

**Published:** 2022-07-28

**Authors:** Yutong Gao, Zhenying Zhang, Binli Chen, Xiying Wang

**Affiliations:** 1School of Education, University of North Carolina at Chapel Hill, Chapel Hill, NC 27599, USA; gyutong@unc.edu; 2Institute for Education Theories, Faculty of Education, Beijing Normal University, Beijing 100875, China; xiyingw@bnu.edu.cn; 3School of Social Development and Public Policy, Beijing Normal University, Beijing 100875, China; blichen@bnu.edu.cn

**Keywords:** adolescent aggression, homophobic teasing/name-calling, gender-based harassment, bullying, gender role attitudes

## Abstract

Homophobic teasing or name-calling, one form of school-related gender-based violence, refers to the use of derogatory language or actions towards sexual- or gender-nonconforming individuals. Research in the Global North has indicated that it is highly prevalent among adolescents, and is associated with a broad range of negative outcomes for both victims and perpetrators. However, such behaviors remain understudied in China. Using a cross-sectional design, the present study investigated the structural relations between homophobic teasing, bullying perpetration, and gender role attitudes among 1915 Chinese high school students. The results showed that 11.5% of the participants had perpetrated such harassment in the past month. Structural equation analyses revealed that bullying perpetration predicted more teasing involvement, and that the relationship was partially mediated by gender role attitudes among both female and male youth. The moderation effect of sex was found only for the direct effect of bullying; such that males who engaged in bullying were more likely to perpetrate homophobic teasing than females. These findings suggest the need for further examination and effective interventions and preventions for the behavior in Chinese contexts.

## 1. Introduction

Adolescence is a critical developmental stage where youth actively engage in identity exploration and relationship formation. Gender and sexual identities gradually crystallize as youth become increasingly aware of their preferred gender expression and sexual attraction [1]. Some youth may start to adopt gender/sexual minority labels. According to the 2015 Chinese Adolescents Health Survey, which involved 123,459 high school students from seven provinces, five percent of participants self-identified as sexual minorities [2]. Despite the large size of the community, LGBT youth remain invisible due to the mixed picture of LGBT rights in China. On the one hand, homosexuality was decriminalized in 1997 [3] and removed from the third version of the Chinese Diagnostic Criteria of Mental Disorders (CCMD-3) [4]. On the other hand, numerous legal obstacles persist, as same-sex marriage is not recognized, and there is no legal protection against LGBT discrimination [5]. Furthermore, biases and hostilities toward this community persist in society [6,7].

Within the school context, LGBT-friendly policies and supporting services are lacking [8]. Moreover, there is limited educational practice against gender-based violence [9]. Particularly in terms of sexuality education, despite increasing efforts in Chinese schools and research, the topic remains a taboo for many educational practitioners [9]. Comprehensive sexuality education, which often incorporates themes of gender and violence prevention, is even more rare in China due to a variety of concerns [9]. Therefore, sexual/gender minority adolescents continue to face considerable challenges at school, including discrimination, peer aggression, and harassment, and fail to get help when needed. This phenomenon clearly violates the Sustainable Development Goal 4 (quality education) and Goal 5 (gender equality) [10], and prevents LGBT youth from fulfilling their potential. 

Homophobic teasing (or name-calling) is one common form of school-related gender-based violence. It refers to any denigratory language (e.g., “fag”, “homo”) or action (e.g., pejorative gestures) that targets sexual-/gender-nonconforming individuals [11,12]. Such harassment can be directed at any individual, including sexual and gender minorities and heterosexual and cisgender youth who do not conform to traditional gender/sexual norms [13]. Extant evidence from the Global North has recognized it as a highly prevalent behavior among adolescents, ranging from 20% to 70% across studies [14,15,16,17]. A recent cross-national study in Europe shows that 54% of LGBTI youth surveyed had been bullied, and 83% reported hearing negative remarks addressed to someone else due to sexual orientation, gender identity, and gender expression [18]. Studies in Chinese societies have also revealed an alarming phenomenon. For example, in Wei and Liu’s online investigation, which surveyed 756 LGBT students in China, 40.7% of respondents reported that they had experienced homophobic name-calling, making it the most frequent victimization among sexual minority students [19].The victimization of such harassment has been linked with school difficulties [20,21], emotional and behavioral problems [2,20,22,23], as well as suicidal ideation [16,19,20]. Moreover, homophobic harassment can escalate to other forms of violence (e.g., sexual violence), which has been observed in several longitudinal studies (e.g., [11]). To effectively address the phenomena, it is imperative to identify its precursors, and examine the mechanisms that contribute to this behavior. The present study, therefore, aims to examine the predictive role of bullying involvement in homophobic teasing through gender role attitudes.

Previous research has identified school bullying as a critical predictor of homophobic name-calling. Bullying, defined as recurring aggression with an imbalance of power between the perpetrator and the victim, involves diverse forms, including physical attacks, verbal abuse, and relational aggression [24,25]. Despite overlaps in definition and shared risk factors, bullying has been distinguished from homophobic teasing, since it does not always involve sexual- or gender-based content [11]. The present study employs the above conceptualization, and defines bullying as non-sexual/gendered aggression. Increasing evidence has provided support for the link between the two behaviors [26,27,28].

Additionally, the association between bullying and homophobic teasing is likely to be mediated by traditional gender role attitudes. Gender role attitudes, sometimes referred to as gender ideology or gender role beliefs, are individual reflections of societal gender norms, i.e., socially expected attributes and behaviors of females and males [29,30].

Individuals who hold traditional gender role attitudes support stereotypical gender attributes (e.g., agency and dominance for males, kindness and submissiveness for females), sexual double standards (e.g., males have greater rights and power over females during dating and sex) [9,31], and heteronormativity (a belief that heterosexuality is the normal or default sexual orientation) [32,33]. Existing evidence has revealed that adolescent bullying perpetration functions as a reinforcer of traditional gender norms [34,35]. As traditional gender norms value rigid gender ideals and heteronormativity, it is not surprising that the endorsement of traditional gender role attitudes predicts more involvement in homophobic teasing [36].

The above evidence indicates that early bullying involvement may predict homophobic teasing through increased gender traditionality. Despite this growing research, the majority of extant studies on the phenomenon come from the Global North, and research on the topic remains limited in China [21,37,38]. The present investigation, therefore, proposes to advance the knowledge of homophobic teasing by investigating its association with bullying perpetration and gender role attitudes among Chinese adolescents. Particularly, the study seeks to answer the following question: does bullying involvement predict homophobic teasing through traditional gender role attitudes?

## 2. Literature Review

### 2.1. Homophobic Teasing, Bullying Perpetration: Direct Association

Proposed by Espelage and colleagues, the Bully-Sexual Violence Pathway Theory holds that bullying perpetration during early adolescence predicts subsequent involvement in homophobic teasing, which then predicts sexual violence perpetration [26,39]. According to the theorists, bullying becomes gradually sexualized during adolescence: with pubertal development, adolescents engage in more cross-sex interactions and become increasingly interested in exploring gender/sexual identities. Like many other behaviors, aggressive behaviors, such as bullying, which were once non-sexual, gradually expand to include sexual and gender content. Therefore, new forms of aggression, such as homophobic teasing, emerge during this stage [26].

Both cross-sectional and longitudinal studies have offered support for the above pathway: an earlier study by Poteat and Espelage [27] examined the occurrence of homophobic language use and its correlates in middle school students. Among the 191 respondents, scores on the homophobic language scale were significantly correlated with bullying scores (r = 0.61 and 0.58 for males and females, respectively), indicating a moderate-to-strong association between general bullying perpetration and homophobic name-calling in both boys and girls [27]. In a similar vein, Poteat and Rivers [28] found that active involvement in school bullying was significantly associated with the use of homophobic epithets. The study, which included 253 high schoolers, investigated bullying participation by assessing participants’ roles in these incidents. Compared with defenders and uninvolved outsiders, the primary bullying role, defined by proactive engagement in bullying, and supportive roles, including reinforcers and assistants, exhibited the most homophobic name-calling [28].

More robust support for the hypothesis comes from longitudinal research on the two behaviors. One such study, based on a sample of 380 U. S. high school students in Illinois, reported the covariation of bullying perpetration and homophobic name-calling over 2 years [40]. Bullying behaviors and homophobic name-calling shared similar patterns, with higher frequencies in one accompanied by more engagement in the other. Another study, which included 1655 students (fifth-to-eighth-graders) in the Midwestern region of the U.S., also explored the trajectory of bullying and homophobic teasing across two years [41]. Through four waves of data collection, the researchers found that a higher prevalence of bullying, measured by both within-person and between-person rates, was predictive of more teasing behaviors. Such an association was held for both female and male students.

Given the above evidence, the present study hypothesizes that:

**H1.** 
*Bullying perpetration is significantly correlated with homophobic teasing; such that more bullying perpetration predicts greater teasing involvement among both female and male adolescents.*


### 2.2. Homophobic Teasing and Bullying Perpetration: Mediation by Gender Role Attitudes

#### 2.2.1. Predicting Traditional Gender Role Attitudes with Bullying Involvement

According to gender and feminist researchers, gendered behaviors and attributes both embody and reinforce existing gender norms [42]. Through such a lens, adolescent bullying is essentially a gendered practice that maintains traditional or hegemonic masculinity and femininity norms [42]. Particularly, physical and verbal bullying among boys reinforces hegemonic masculinity, which values dominance and violence [43]. Relational bullying among girls enhances hegemonic femininity, which endorses rivalry between female peers [44]. 

However, limited empirical evidence exists regarding the effect of bullying on gender role attitudes, except several qualitative investigations; through which, participants reflected on the meaning of their bullying involvement [42]. One such study included 275 middle school boys from 12 states in the U.S., and investigated participants’ school victimization experiences [35]. Through content analysis of students’ open-ended responses, the investigators found that bullying victims acknowledged the strong, dominant, and violent masculine ideal that bullying perpetrators demonstrated. For instance, many respondents mentioned physical attacks or intimidation by aggressors, and indicated their willingness to retaliate. Most victims shrugged off their victimization experience, often remarking, “boys will be boys” [35]. These responses, as the researchers reasoned, reflected that bullying behaviors substantiated traditional male gender roles, particularly toxic masculine norms of aggression.

Similarly, in Carrera-Fernández and colleagues’ study [34], which interviewed 93 Spanish adolescents through 12 focus group sessions, bullying experience among girls, particularly the exclusion of “indecently dressed” peers, served to reproduce female rivalry and the “virgin/whore” discourse in hegemonic femininity. As the researchers pointed out, such experience sustained the need for evaluation and surveillance of themselves and their peers, contributing to the traditional female gender roles, particularly the competitive and pure feminine ideal.

Despite this qualitative evidence, the majority of research on bullying and gender role attitudes focuses more on the predictive role of gender role attitudes on bullying than vice versa. Therefore, the present study extends existing research by exploring the contribution of bullying behaviors on gender role attitudes.

#### 2.2.2. Predicting Homophobic Teasing with Traditional Gender Role Attitudes

The Gender Role Strain Paradigm [45] construes the pressure of conforming to dominant gender ideologies as the source of problem behaviors among both males and females. According to Pleck, traditional (or hegemonic) gender ideologies can be dysfunctional. For example, adherence to masculine ideologies demands heteronormativity. When exalted to the point of legitimacy (e.g., in terms of sexuality, traditional masculine ideologies conceive heterosexuality as the only legitimate orientation), these traits become problematic, and, thus, predispose people to behavioral problems, such as violence towards sexual minorities. In this way, homophobic teasing results from the pressure to conform to traditional masculine norms. Individuals who perpetrate such harassment use it to assert their masculinity.

Consistent with the above hypothesis, increasing evidence has been found regarding the impact of traditional gender role beliefs on homophobic teasing and related behaviors. One earlier meta-analysis, which examined the relationship between gender role-related constructs and homophobic attitudes and behaviors, concluded that the endorsement of masculine norms and conservative gender attitudes were strong predictors of such attitudes and behaviors, although the strength of the association varied across studies [46]. Recent studies have afforded more support for the finding. For instance, in Valido and colleagues’ longitudinal investigation of homophobic name-calling based on the same sample mentioned above [41], high between-person endorsement of traditional masculinities was one of the significant risk factors for teasing perpetration. Research from other countries, including Spain, Portugal [47], and Switzerland [48], has revealed similar connections.

Accordingly, the present study hypothesizes:

**H2.** 
*The association between bullying perpetration and homophobic teasing is mediated by traditional gender role attitudes, such that bullying behaviors contribute to teasing perpetration through the indirect effect of traditional gender role attitudes.*


### 2.3. Sex Differences in the Impact of Bullying on Homophobic Teasing

Though bullying predicts homophobic teasing, the strength of this relationship appears to differ by sex. Several studies have noted this difference. For instance, in Poteat and Espelage’s study mentioned above [27], males with relational bullying involvement were more likely to tease others than females, although the difference was not statistically significant. In a similar vein, the more recent investigation by Poteat and Rivers [28] reported that male youth, but not females, who engaged in multiple bullying roles were more likely to also be involved in homophobic name-calling.

Given the above research, the present study intends to explore sex differences within the two proposed pathways, bullying perpetration and teasing involvement. Specifically, the study hypothesizes that: 

**H3.** 
*Sex moderates both the direct and indirect relationship between bullying perpetration and homophobic teasing, such that male bullying perpetrators are more likely to be homophobic teasers. Moreover, the mediation effect through gender role attitudes is stronger among males.*


### 2.4. Other Factors Related to Bullying, Gender Role Attitudes, and Homophobic Teasing

A set of sociodemographic factors, including age, birthplace, parental educational level, and socioeconomic statuses, have also been implicated in the key variables. For instance, much evidence suggests that both bullying and homophobic teasing decline with age, possibly due to growing cognitive capabilities that restrain violent behaviors and reject prejudice [49]. Moreover, youth from urban (versus rural) areas and better family socioeconomic conditions, and whose parents received more education, have been shown to have more equitable gender role attitudes [50]. Therefore, the above variables were controlled in the present research to account for potential confounding effects.

## 3. Materials and Methods

### 3.1. Procedure

The data in the present study were collected in a large-scale survey on Chinese adolescent psychosocial functioning that was conducted in 2018. The original survey was approved by the Institutional Review Board of Beijing Normal University, and carried out by several faculty members and graduate students from the institution. After locating two cities of interest, i.e., one in the western region of China and the other on the eastern coast, the research team selected eleven schools, including middle schools, high schools, and vocational schools in both urban and rural regions from the two cities. After getting approval from the local educational bureau, the principals of each school were approached and asked to sign a letter of support if they agreed to participate in the study. Once a school agreed to participate, four to five 8th- and 10th-grade classes within the school were randomly selected. All students in chosen classes were eligible to participate. Researchers helped explain the nature and procedure of the survey, and informed students that they had the right to decline participation without any penalty. Printed forms of student consent were distributed. Due to the minimal risks presented by the study, parental consent was waived. Students were asked to sign the form before they participated. All students who signed consent forms completed the survey questionnaire in their own classrooms, which took approximately 40 min. During the survey, one researcher was present in each classroom, but kept some distance from the students (e.g., sitting in the front of the classroom); students were told that they could seek help from the researcher if necessary. All questionnaires were completed under anonymity, with no identifiable private information, except a pseudo-ID number. All respondents were given small gifts (e.g., stationary worth approximately 0.5 to 1 dollars) to thank them for their participation. A total of 4000 questionnaires were distributed, and 3531 questionnaires were collected, with a response rate of 88.3%. In the present study, only the responses of high school students were included. The reason is that, compared with younger counterparts, adolescents aged 15 to 18 engage in more dating behaviors and become more sexually active. Their increased interest in gender and sexuality issues is likely accompanied by greater risks of violence against individuals that do not conform to traditional standards.

### 3.2. Sample

The present sample comprised 1915 Chinese high school students who participated in a large-scale psychosocial survey conducted in July 2018. The survey employed a convenient sample from two cities in China, Chongqing and Suqian. Chongqing, located in Western China, is a large city with approximately 30 million people. Suqian, a mid-sized city in Jiangsu Province, is located in Eastern China, with a population of 4.9 million people. Across two cities, eleven schools in total were recruited, including middle schools, high schools, and vocational schools from both urban and rural areas. A detailed description of sample characteristics is shown in Table 1.

The average age of participants was 15.73 years (SD = 0.908, range = 13–20), with an approximately even proportion of males (50.7%) and females (49.3%). In terms of hukou, a residential system in China that approximates the birthplace, more students were from urban regions (65.8%) than rural (34.3%). Overall, most respondents’ fathers (59.2%) and mothers (63.9%) did not attend college or complete college education. Thirty-five percent of fathers and twenty-nine percent of mothers had a bachelor’s degree or higher. Over 70.6% of participants perceived their families to be average compared with their peers in terms of their socioeconomic conditions.

### 3.3. Measures

#### 3.3.1. Homophobic Teasing

Homophobic teasing was assessed using one item which asked respondents whether they taunted or name-called other students in the past month. A 4-point scale was used to reflect the frequency of such harassment (never = 0, once or twice = 1, three or four times = 2, five or six times = 3, seven times or more = 4). The average score among 1912 complete cases was 0.17 (SD = 0.556, range = 0–4).

#### 3.3.2. Bullying Perpetration

Bullying perpetration was measured with the Illinois Bullying Scale, a widely used bullying assessment tool with sufficient psychometric evidence [51]. The original scale has 18 items across three subscales, i.e., the bully subscale, the fight subscale, and the victim scale. The bully scale contains nine items that cover various bullying behaviors, for example, “I upset other students for the fun of it”, “I started arguments or conflicts”, and “I excluded others.” All items refer to the month before the survey, and were followed by the same response categories as homophobic teasing (from never = 0 to 7 times or more = 4). The scale has been used in Chinese samples, and demonstrated good internal consistency [52]. Cronbach’s alpha in the present sample was 0.804. There were 1897 complete cases, with a mean score of 2.045 (SD = 0.448, range = 0–4).

#### 3.3.3. Gender Role Attitudes

Gender role attitudes were assessed using the Attitudes Towards Women Scale for Adolescents [53]. The scale, including 12 items (e.g., “swearing is worse for a girl than for a boy” and “in general, the father should have greater authority than the mother in making family decisions”), has been extensively used across nations, and was shown to be psychometrically sound [54]. Each response was scored from “strongly agree” (1 point) to “strongly disagree” (4 points). A total score across the 12 items was summed to indicate a global attitude. A high score indicates a more egalitarian attitude. Similar forward and backward translation procedures with the bullying scale were performed on the scale, followed by explanatory and confirmatory factor analysis to examine its utility in Chinese samples. Based on the results, two items (item 2, “On a date, the boy should be expected to pay all expenses”, and item 7, “It is all right for a girl to ask a boy out on a date”) were eliminated due to consistent low loadings in factor analyses (<0.3) and compromises on internal consistency. This issue, as the authors reason, may be related to the cultural difference underlying the two items; that is, because of strict prohibitions on dating in Chinese middle and high schools, participants in the present sample may find it hard to relate to the scenario portrayed by the items. The resulting scale had ten items, and Cronbach’s alpha was 0.63. Among 1906 complete cases, the average score was 2.089 (SD = 0.447, range = 0–4).

#### 3.3.4. Control Variables

Based on the extant findings, a series of control variables were also included. Age, sex (female = 0, male = 1), hukou (rural = 0, urban = 1), paternal and maternal educational level (from no formal education = 1, bachelor’s degree or higher = 6, to others = 9), and socioeconomic status (from much better than peers = 1, average = 3, to much worse than peers = 5) were entered into the model.

### 3.4. Data Analysis

All statistical analyses were conducted using SPSS 26.0. Preliminary analyses were first performed to examine the descriptive statistics and correlations between variables. Structural equation modeling was then used to examine the hypothesized relationships with a bootstrap sampling method. In the invariance testing of the structural model, multiple regression was used to test the hypotheses. Missing values were estimated using Maximum Likelihood Estimate in SPSS.

Within the structural model, two regression equations were analyzed, regressing homophobic teasing and gender role attitudes on bullying perpetration, respectively. The mediating effect of gender role attitudes was assessed following the procedure in MacKinnon and colleagues’ study involving bootstrap analyses. The reason for using this method is that bootstrapping does not need to assume the normal distribution of samples. Instead, it estimates the indirect effect and bias-corrected confidence intervals through resampling procedures [55]. In the present study, bootstrapping with 5000 resamplings was employed using the SPSS macros function developed by Preacher and Hayes [56]. For the results, 95% confidence intervals were checked; the indirect effect is significant if the CIs do not include zero [55]. Finally, the moderation effect of sex was checked using the interaction term of bullying*sex and gender role attitude*sex.

## 4. Results

### 4.1. Descriptive Statistics of Key Variables

#### 4.1.1. Prevalence of Bullying Perpetration

In the current sample, i.e., 958 boys and 937 girls, the average bullying behavior score was 0.377 (SD = 0.518). The average score of bullying behavior among boys was 0.442 (SD = 0.606), and that of girls was 0.31 (SD = 0.4), showing a significant gap between males and females (t = 5.625, *p* < 0.001); that is, males perpetrated more bullying than girls.

#### 4.1.2. Average Gender Role Attitudes

Among 958 boys and 937 girls, their average gender role attitudes score was 2.089 (SD = 0.447). On average, boys scored 1.88 (SD = 0.399) and girls scored 2.30 (SD = 0.394), indicating more equitable gender attitudes among girls than boys (t = −22.779, *p* < 0.001).

#### 4.1.3. Prevalence of Homophobic Teasing Perpetration

The rate of homophobic teasing in the present sample was 11.5 percent. Specifically, among 1912 high school students, 219 students had teased others, including 167 (76.3%) who did it once or twice, 24 (10.9%) for three to four times, 11 (5.02%) for five to six times, and 17 (7.76%) for more than seven times. The average score for homophobic teasing was 0.24 (SD = 0.69) for boys and 0.08 (SD = 0.34) for girls, respectively, with a mean of 0.17 for all students. Therefore, the perpetration rate among male students was significantly higher than females (t = −6.125, *p* < 0.001).

### 4.2. Correlations between Variables

Table 2 presents Pearson correlations among the variables. As expected, there was a significant positive relationship between bullying and homophobic teasing. Gender role attitudes had a small negative correlation with bullying, and a moderate negative correlation with homophobic teasing.

### 4.3. Structural Relations: Invariance

Within the structural model, multiple linear regression analysis was used to test hypotheses 1, 2, and 3. In each model, control variables (age, sex, parental education level, and family socioeconomic status) were entered, followed by predictor variables. The results are presented in Table 3. 

Model 2 shows that bullying perpetration could predict homophobic teasing (B = 0.776, *p* < 0.01), which supported hypothesis 1. There was a negative relationship between gender role attitudes and homophobic teasing (B = −0.161, *p* < 0.01). In model 1, bullying perpetration was negatively associated with gender role attitudes (B = −0.201, *p* < 0.01). Additionally, the results show that the indirect effect of gender role attitudes through the bullying–homophobic-teasing path (a*b = 0.032, *p* < 0.01, CI = (0.015,0.058)) was significant, and none of the bias-corrected CI contained zero. Therefore, hypothesis 2 was supported.

Finally, we examined the moderating effect (hypothesis 3) by referring to the steps of Preacher, Rucker, and Hayes’ study [57] in testing the indirect effects of conditions and development macros. In the case of only one mediating variable, the conditional indirect effect represents the size of the indirect effect at the specific level of the moderating variable [57,58]. Accordingly, multiple regression was conducted by regressing homophobic teasing first on bullying and the interaction term of bullying and sex, and then on gender attitudes and their interaction term with sex. The results are shown in Table 3. Compared with model 1, model 2 showed a significant cross-term effect between bullying*sex (B = 0.502, CI = (0.240,0.765), *p* < 0.01). Then, we set the Bootstrap sample to 5000 times, and ran the macro to test the indirect effect of the condition. Across sex, the relationship between bullying and homophobic teasing remained significant, indicating the existence of a moderating effect. However, there was no moderating effect on the path from bullying perpetration to gender role attitudes or from gender role attitudes to teasing. The above results, therefore, partially support hypothesis 3; that is, the relationship between bullying and homophobic teasing was stronger in males than in females (male = 1.268, *p* < 0.01; female = 0.766, *p* < 0.01). Figure 1 shows the structural relations.

## 5. Discussions

The present investigation extended previous research on homophobic teasing by examining the behavior among an understudied sample, i.e., Chinese adolescents. The results showed that 11.5% of high school students in the sample had perpetrated homophobic teasing in the past month. The rate, although it suggests the commonness of the behavior, is lower than those reported in other studies (e.g., [16,21]). Such a difference could indicate sample heterogeneity; that is, adolescents in the present sample conducted less teasing than those in other studies. However, it may also be attributed to methodological discrepancies, for example, differential screening tools for homophobic teasing. In the present study, only one item was used for assessment, which could have underestimated the occurrence of the behavior. In addition, the study found that boys engaged in more bullying behaviors than girls, which is consistent with previous studies [59,60,61]. For gender role attitudes, scores on the scale indicated that girls held more equitable attitudes than boys, which is also in line with the previous literature [62,63,64,65]. With the control variables, echoing existing evidence, being younger, having an urban background, and a higher maternal educational level were related to more equitable gender role attitudes [63]. 

In terms of the association between bullying and homophobic teasing, the present study found that bullying perpetration was related to higher odds of engaging in teasing behaviors. This finding is in line with extant research [11,28,39], indicating that similar patterns between the two forms of aggression may also exist in Chinese contexts. Further, the study suggested that gender role attitudes partially mediated the pathway from bullying to homophobic teasing. Therefore, the findings support the proposition that bullying behaviors reproduce traditional gender norms, predisposing perpetrators to teasing involvement. 

In addition to the above findings, the present study also found a significant moderating effect of sex on the direct pathway from bullying to teasing involvement. In particular, males who engaged in bullying were more likely to also be involved in homophobic teasing than females. This finding is consistent with existing evidence on the differential impact of bullying across sex [28]. However, such effects were not observed for the indirect path through gender role attitudes. 

### 5.1. Limitations and Directions for Future Studies

Several limitations of the present study are worth noting. First, the assessment tool of homophobic teasing, which was self-reported and included only one item, could have underestimated the prevalence of the behavior. However, the lack of appropriate screening tools is largely related to the limited research on the phenomena in China. Future research is, therefore, encouraged to validate existing tools or develop new tools that can be utilized in Chinese contexts. Second, given that the data were collected from two cities in China using convenient sampling, the generalization of the present findings to the entire nation should be cautioned. For a clearer picture of the phenomena, nationally representative samples through random sampling are needed. Third, the cross-sectional nature of the present study precludes any causal inference between the study variables. For future research, longitudinal designs should be adopted, which helps to elucidate the relationships between the factors.

### 5.2. Implications for Practice

The present study has several implications for educators and policy makers. Overall, given the commonness of homophobic teasing in the present sample, intervention and prevention efforts addressing the behavior are in urgent need. Notably, the association between bullying, gender role attitudes, and homophobic teasing indicates that effective intervention of teasing behaviors should also address bullying involvement and promote gender egalitarianism. Within school contexts, it is imperative that both anti-discrimination and bullying prevention policies be established to prohibit the related harassment. Furthermore, it is suggested that comprehensive violence prevention and sexuality education programs be conducted, which address bullying behaviors, LGBT-related biases, as well as inequitable gender role attitudes. Such programs should involve multiple stakeholders through diverse approaches, for example, curriculum and activities for students, and training workshops for teachers and school administrators. In addition, given that male bullies are more likely to become teasers than females, the intervention approaches mentioned above should pay particular attention to boys by focusing on teaching them about the negative consequences of toxic hegemonic masculinity, and promoting healthy masculinities.

## 6. Conclusions

This study is significant in the following ways: first, this is one of the few studies focusing on homophobic teasing among Chinese adolescents, which contributes to the understanding of the relations between bullying, homophobic teasing, and traditional gender roles. Second, the study suggests that anti-homophobic teasing needs to be an essential part of both bullying intervention and comprehensive sexuality education. The revised International Technical Guidance on sexuality education included two new concepts, key concept 3 (Understanding Gender) and key concept 4 (Violence and Staying Safe) [32]. This study highlights the significance of both concepts, especially the importance of the promotion of students’ understanding of SOGIE (sexual orientation, gender identity, and gender expression), and facilitating critical reflections on gender roles, norms, stereotypes, sexual double standards, and heteronormativity. This study also shows it is critical for students to take active actions to advocate for safe school environments. Third, this study echoes the 2030 Sustainable Development Goal 4 (equitable and inclusive education) and Goal 5 (gender equality). Protecting sexual- and gender-non-conforming learners from homophobic and other gender-based bullying and violence is essential to creating a positive and supportive school environment. 

## Figures and Tables

**Figure 1 children-09-01127-f001:**
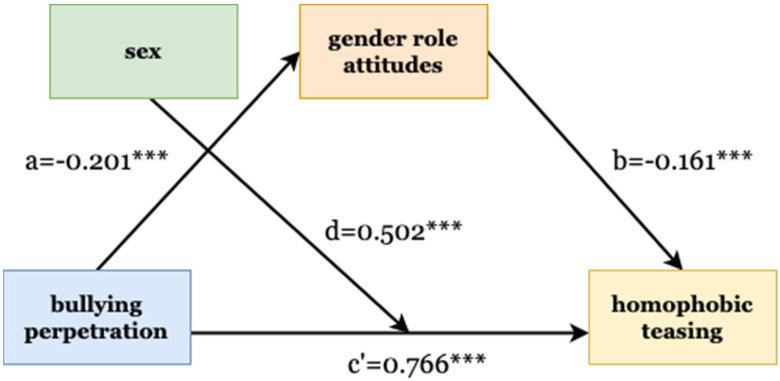
Structural model of relations between variables. *** = *p* < 0.01.

**Table 1 children-09-01127-t001:** Participants’ demographic characteristics.

Demographic Variables	Sex	Mean ± SD
Sex	(Female:Male)	1:1.028
Age	Male	15.88 ± 0.977
	Female	15.57 ± 0.802
Hukou (birthplace)	Male	0.658 ± 0.474
	Female	0.658 ± 0.475
Maternal educational level	Male	4.01 + 1.813
	Female	4.12 + 1.763
Paternal educational level	Male	4.34 + 1.685
	Female	4.42 + 1.636
Socioecomonic status	Male	3.05 ± 0.0766
	Female	3.02 + 0.611

**Table 2 children-09-01127-t002:** Pearson correlations between variables.

Variables	1	2	3
1. Homophobic teasing			
2. Bullying perpetration	0.410 **		
3. Gender role attitudes	−0.205 **	−0.091 **	

** = *p* < 0.05.

**Table 3 children-09-01127-t003:** Indirect and total effects between variables (standardized values).

	Gender Role Attitudes	Homophobic Teasing
Variables	Model 1	Model 2
Age	−0.054 ***	0.05
Hukou	0.129 ***	−0.007
Maternal educational level	0.013 *	−0.012
Paternal educational level	−0.002	0.008
Socioeconomic status	0.016	−0.010
Bullying perpetration	−0.201 ***	0.766 ***
Sex	−0.389 ***	−0.049
Bullying perpetration*sex	−0.016	0.502 ***
Gender role attitudes		−0.161 ***
Gender role attitudes*sex		−0.042
R-square	0.075	0.193

*** = *p* < 0.01, * = *p* < 0.1.

## Data Availability

The data are not publicly available due to privacy and ethical concerns.

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
