# Peer review of "Bullying Perpetration and Homophobic Teasing: Mediation through Gender Role Attitudes"

_children, 2022, doi:10.3390/children9081127_

Round 1

Reviewer 1 Report

This is an excellent article. It brings attention to homophobic teasing and bullying in a large sample of Chinese  high school students. It isn't a surprise that males engage in these behaviors more than females, but this is an important addition to the literature. A strength of this work is the clarity with which it is written.

The abstract is very good. Terms such as "homophobic teasing" are well-defined. The literature review is very strong. The sample size is excellent. The hypotheses are clear.

Regarding the article title, "The" before the word "Mediation" can be removed.

I am not a statistician and hope that another reviewer can can be helpful here.

Line #223 states that this study includes only the data from high school students. I suggest some explanation be added here. Is it because there was an issue with data from middle and vocational students? Is it because these other data are the subject of a separate article? It is may not be necessary to repeat the information in line #s 229-230.

The authors have combined the Discussion and Conclusions into one section. It is only a personal preference, but I prefer to see them separated and, of course, this is something for the authors to decide. I would simply place the word "Conclusion" as a heading before the final paragraph. To me, it makes the conclusion appear stronger.

Reviewer 2 Report

Dear Editor and authors,

Firstly, I thank you for my consideration as a reviewer of this manuscript. It is my pleasure to contribute to Children.

This manuscript provides evidence on the structural relations between homophobic tesing, bullying perpetration, and gender role attitudes.

Some issues that they should consider addressing before publication are described:

In the Introduction section, I recommend including information about the sexual double standard in adolescents such as a traditional gender role attitude. Moreover, the authors would include the complete redaction of the third hypothesis.

In the procedure section, the name of the city or the region or the characteristic of the city (e.g., population size) could be included in the manuscript.  Also, information about adult consent for adolescent participation is necessary to include in the manuscript. If the participants were compensated, this information would be included in this section. I would like to ask the authors if the study had inclusion or exclusion criteria.

Due to the analysis examining the sex effect, more information about the sociodemographic characteristics by sex should be included.

 I recommend including the items for evaluation homophobic testing.

The authors could describe why they used control variables in the introduction section and data analysis.

In the data analysis section, it is necessary to include information about the treatment of missing data.

In the results section, I suggest also including the prevalence of bullying perpetration and gender role attitudes. Figure 1 should be indicated in the text.

In general, the discussion section needs more arguments. For example, I suggest discussing more the relationship between bullying and gender role attitudes with previous evidence, the effect of sex in the bullying, and the moderating effect. In addition, the authors did not mention the role of control variables (age, sex, parental education level, and family socioeconomic status) in the regression models.

Finally, more limitations about the collection and characteristics of the sample should be mentioned. I recommend indicating evidence for supporting the conclusion.

Round 2

Reviewer 2 Report

I would like to thank the editor for considering me as a reviewer and the authors for their research. I confirm that the authors have made the suggested modifications.